# A Novel Approach to Parameter Determination of the Continuous Spontaneous Localization Collapse Model

**DOI:** 10.3390/e25020295

**Published:** 2023-02-04

**Authors:** Kristian Piscicchia, Alessio Porcelli, Angelo Bassi, Massimiliano Bazzi, Mario Bragadireanu, Michael Cargnelli, Alberto Clozza, Luca De Paolis, Raffaele Del Grande, Maaneli Derakhshani, Diósi Lajos, Sandro Donadi, Carlo Guaraldo, Mihai Iliescu, Matthias Laubenstein, Simone Manti, Johann Marton, Marco Miliucci, Fabrizio Napolitano, Alessandro Scordo, Francesco Sgaramella, Diana Laura Sirghi, Florin Sirghi, Oton Vazquez Doce, Johann Zmeskal, Catalina Curceanu

**Affiliations:** 1Centro Ricerche Enrico Fermi—Museo Storico della Fisica e Centro Studi e Ricerche “Enrico Fermi”, 00184 Rome, Italy; 2Laboratori Nazionali di Frascati, INFN, 00044 Frascati, Italy; 3Department of Physics, University of Trieste, 34127 Trieste, Italy; 4Section of Trieste, Istituto Nazionale di Fisica Nucleare, 34149 Trieste, Italy; 5IFIN-HH, Institutul National pentru Fizica si Inginerie Nucleara Horia Hulubei, 077125 Măgurele, Romania; 6Stefan-Meyer-Institute for Subatomic Physics, Austrian Academy of Science, 1030 Wien, Austria; 7Excellence Cluster Universe, Technische Universität München, 80333 München, Germany; 8Department of Mathematics, Rutgers University, New Brunswick, NJ 08854, USA; 9Department of Physics of Complex Systems, Eötvös Loránd University, 1117 Budapest, Hungary; 10Institute for Solid State Physics and Optics, Wigner Research Centre for Physics, 1525 Budapest, Hungary; 11Laboratori Nazionali del Gran Sasso, INFN, 67100 L’Aquila, Italy

**Keywords:** collapse models, CSL, spontaneous radiation, germanium detectors

## Abstract

Models of dynamical wave function collapse consistently describe the breakdown of the quantum superposition with the growing mass of the system by introducing non-linear and stochastic modifications to the standard Schrödinger dynamics. Among them, Continuous Spontaneous Localization (CSL) was extensively investigated both theoretically and experimentally. Measurable consequences of the collapse phenomenon depend on different combinations of the phenomenological parameters of the model—the strength λ and the correlation length $r_{C}$—and have led, so far, to the exclusion of regions of the admissible ($\lambda -r_{C}$) parameters space. We developed a novel approach to disentangle the λ and $r_{C}$ probability density functions, which discloses a more profound statistical insight.

## 1. Introduction

The mechanism at the basis of the transition from Quantum to Classical behavior is not explained in the original Quantum Theory (QT) and has puzzled the scientific community since its inception. The superposition principle is a trademark of QT, accounting for many phenomena which cannot find a counterpart in Classical Mechanics. It is a consequence of the linearity of the Schrödinger equation, which has to break down at a certain scale to avoid preposterous predictions concerning the macroscopic bodies’ dynamics.

For several decades, phenomenological dynamical models of wave function collapse have been developed (see, e.g., [1,2,3,4,5,6]; for a review and references, see also [7]), which explain the quantum-to-classical transition by a progressive reduction of the superposition, proportional to the increase in mass of the system under consideration.

Technological developments have recently paved the way for bringing this issue into experimental investigations. Several techniques are presently being employed, constraining the phenomenological parameters introduced in the collapse dynamics (see, e.g., Ref. [8] for a review on the subject). Interferometric experiments deal with the measurement of the interference pattern of the spatial superposition, which is created in an interferometer (see, e.g., [9,10,11,12,13]). Indirect tests of the collapse mechanism can also be performed with non-interferometric experiments, exploiting the unavoidable random motion related to the system interaction with the collapsing field. These types of experiments involve cold atoms [14], optomechanical systems [15,16,17,18,19,20,21], phonon excitations in crystals, [22,23], gravitational wave detectors [24], X-ray and γ-ray measurements [25,26,27,28]. Non-interferometric experiments can probe the effect of the collapse process on macroscopic objects, thus leading, thanks to the amplification mechanism, to extremely sensitive bounds. That is the case of X and γ-rays measurements, which set the strongest constraints over broad ranges of the typical collapse model parameters’ space.

Continuous Spontaneous Localization (CSL) is one of the better-investigated dynamical reduction models. CSL consists of a non-linear and stochastic modification of the Schrödinger equation; non-linearity is needed to suppress quantum superposition and stochasticity to avoid faster-than-light signaling and recover the Born rule [7]. The dynamics is characterized by the interaction with a continuous set of independent noises (one for each point of the space) having, under the simplest assumption, a null average and white correlation in time. The new stochastic terms require introducing two phenomenological quantities: a collapse rate λ, which sets the strength of the collapse, and a noise correlation length $r_{C}$, which measures the spatial resolution of the collapse. Various theoretical considerations lead to different choices for the parameters: Ghirardi, Rimini, and Weber [3] proposed $\lambda =10^{-17}$ s${}^{-1}$ and $r_{C}=10^{-7}$ m, while Adler [29] proposed $\lambda =10^{-8\pm 2}$ s${}^{-1}$ when $r_{C}=10^{-7}$ m and $\lambda =10^{-6\pm 2}$ s${}^{-1}$ when $r_{C}=10^{-6}$ m.

In addition to causing the collapse of the wave function in space, the interaction with the stochastic noise induces a diffusion in space, resulting in a Brownian-like motion. For a system of charged particles, this Brownian-like diffusion causes the particles to emit radiation, known as *spontaneous radiation.* The standard QT does not include such a phenomenon. The noise-induced radiation emission can then be used to test the collapse models.

The spontaneous radiation rate for the CSL model was calculated for an atomic system in Ref. [27]:(1)$${\left.\frac{d\Gamma }{dE}\right|}_{t}=N_{atoms}\cdot \left(N_{A}^{2}+N_{A}\right)\cdot \frac{\hbar \;e^{2}}{4\;\pi^{2}\;\epsilon_{0}\;c^{3}\;m_{0}^{2}}\cdot \frac{\lambda }{r_{C}^{2}}\frac{1}{E}\;,$$
where $N_{atoms}$ is the number of atoms in the system with the atomic number $N_{A}$, *c* is the speed of light, $\epsilon_{0}$ is the vacuum permittivity, $m_{0}$ is the nucleon mass, *E* represents the energy, and *t*—the time. Equation (Equation 1) is exact in the high-energy γ-ray domain (where relativistic electrons’ contribution has to be ignored, and the bracket is reduced to $\left(N_{A}^{2}\right)$). The generalization of this equation to the low-energy range is presently under theoretical investigation. Cancellation effects are expected when the photons’ wavelengths become comparable to the dimensions of the atomic orbits.

The experimental studies of the spontaneous radiation phenomenon focused so far on the $\lambda /r_{C}^{2}$ ratio, which regulates the predicted yield. The strongest limits from γ-rays ($\lambda <52\;r_{C}^{2}$ s${}^{-1}$ [27]) and X-rays ($\lambda <0.494\pm 0.015\;r_{C}^{2}$ s${}^{-1}$ [28]) allow to exclude regions of the ($\lambda -r_{C}$) parameter space. The combined information from theoretical considerations [30] and other experiments [24] has led to the further exclusion of sectors of the ($\lambda -r_{C}$) plane, characterized by a different functional relation between λ and $r_{C}$. Including this rich prior information in the statistical analysis permits to disentangle the two parameters’ probability density functions (pdfs). The individual pdfs disclose a much deeper insight into the state-of-the-art knowledge of the strength and correlation length of the model.

This work aims to provide an analytic method to extract the *pdf*s of the λ and $r_{C}$ parameters utilizing a Bayesian comparison of the measured spectrum with the expected spontaneous radiation contribution Equation (Equation 1) and the simulated background. Such a procedure cannot be currently, consistently, applied to the data presented in Ref. [28], since an absolute background yield for this measurement is not provided, the background contribution being instead inferred from the data fit. To provide an example of the application of this novel technique, the data set recently measured in Ref. [27] will be re-analyzed in the new framework, exploiting as prior, for the spontaneous radiation rate, the previous result [25]. The analysis is performed in the energy range ΔE=(1−3.8) MeV, where cancellation corrections to Equation (Equation 1) are expected to be negligible.

## 2. The Experimental Setup

The experiment was operated in the Gran Sasso underground National Laboratory of INFN, where the overburden of the Gran Sasso mountain, corresponding to a minimum thickness of 3500 m w.e. (metres water equivalent), provides the ideal environment for the search of the spontaneous radiation emission. The background, aside from the residual cosmic rays, is mainly produced by long-lived γ-emitting primordial isotopes and their decay products.

The measurement was performed with a coaxial p-type High Purity Germanium detector (1.982 kg in mass), surrounded by a 62 kg sample of electropolished oxygen-free high-conductivity copper in Marinelli geometry and enclosed in a shielding structure made of an external pure lead layer (30 cm from the bottom and 25 cm from the sides) and an inner 5 cm thick electrolytic copper layer. Shielding and cryostat are contained in an air-tight steel housing, flushed with boil-off nitrogen to reduce radon contamination. Additionally, 5 cm thick borated polyethylene plates are placed on the bottom and the sides, which partially reduces the neutron flux going toward the detector. The experimental setup is schematically shown in Figure 2 of Ref. [26]; more details on the shielding, the cryogenic, and the vacuum systems are given in Refs. [31,32]. The data acquisition system is a Lynx digital signal analyzer controlled via GENIE 2000 personal computer software, both from Canberra-Mirion.

The measured emission spectrum corresponds to a data-taking period Δt of about 62 days and is shown in black in Figure 1 of Ref. [27], in the analyzed range ΔE.

## 3. Joint Probability Distribution Function of λ and $R_{c}$

The total number of counts in ΔE ($z_{c}=576$) follows a Poissonian distribution:(2)$$p\left(z_{c}|\Lambda_{c}\right)=\frac{\Lambda_{c}^{z_{c}}e^{-\Lambda_{c}}}{z_{c}!}.$$

The expected number of total counts ($\Lambda_{c}$) can be expressed in terms of the expected signal ($\Lambda_{s}$) and background ($\Lambda_{b}$) contributions:(3)$$\Lambda_{c}=\Lambda_{b}+\Lambda_{s}\left(\frac{\lambda }{r_{C}^{2}}\right).$$

The total number of spontaneously emitted photons, which would be detected during the acquisition time Δt, is obtained by weighting the theoretical rate with the efficiency functions and summing over the setup constituents (*i*):(4)$$z_{s}\left(\frac{\lambda }{r_{C}^{2}}\right)=\sum_{i}\int_{\Delta E}{\left.\frac{d\Gamma }{dE}\right|}_{t}^{i}\;\Delta t\;\epsilon_{i}\left(E\right)\;dE=2.0986\frac{\lambda }{r_{C}^{2}}=a\frac{\lambda }{r_{C}^{2}}\;,$$
and $\Lambda_{s}=z_{s}+1$. The efficiency spectra for the apparatus components giving a detectable contribution are shown in Figure 2 of Ref. [27], and the parameters of the corresponding fit functions are summarized in Table 1 of the same paper. The experimental setup was completely characterized through a validated Monte Carlo (MC) code [33] (based on the GEANT4 software library, Ref. [34]), which was used to produce the efficiency spectra and the background simulation. The measurements of the activities of the radionuclides of each part of the setup served as input of the background MC, which considers the emission probabilities, the decay schemes, the photons’ propagation and interactions, and the detection efficiencies. The simulated background spectrum is shown in Figure 1 of Ref. [27] in magenta. The expected number of background counts in ΔE is found to be:(5)$$\Lambda_{b}=507.$$

Making explicit in the *pdf* of $z_{c}$ (Equation (Equation 2)) the dependence on the parameters λ and $r_{C}$, we have
(6)$$p\left(z_{c}|\lambda ,r_{C}\right)=\frac{{\left(a\frac{\lambda }{r_{c}^{2}}+\Lambda_{b}+1\right)}^{z_{c}}e^{-\left(a\frac{\lambda }{r_{c}^{2}}+\Lambda_{b}+1\right)}}{z_{c}!}\;.$$

Therefore, the joint pdf of λ and $r_{C}$ is obtained by applying the Bayes formula for multi-dimensional continuous distributions:$$\tilde{p}\left(\lambda ,r_{C}|z_{c}\right)=\frac{p\left(z_{c}|\lambda ,r_{C}\right)\cdot {\tilde{p}}_{0}\left(\lambda ,r_{C}\right)}{\int_{D_{\lambda ,r_{C}}}\;p\left(z_{c}|\lambda ,r_{C}\right)\;dr_{c}\;d\lambda }$$
(7)$$\Rightarrow \tilde{p}\left(\lambda ,r_{C}\right)=\frac{{\left(a\frac{\lambda }{r_{c}^{2}}+\Lambda_{b}+1\right)}^{z_{c}}e^{-\left(a\frac{\lambda }{r_{c}^{2}}+\Lambda_{b}+1\right)}\cdot {\tilde{p}}_{0}\left(\lambda ,r_{C}\right)}{\int_{D_{\lambda ,r_{C}}}{\left(a\frac{\lambda }{r_{c}^{2}}+\Lambda_{b}+1\right)}^{z_{c}}e^{-\left(a\frac{\lambda }{r_{c}^{2}}+\Lambda_{b}+1\right)}\;dr_{c}\;d\lambda }.$$

The prior ${\tilde{p}}_{0}\left(\lambda ,r_{C}\right)$ is chosen to reduce the domain ($D_{\lambda ,r_{C}}$) of the stochastic variables $\left(\lambda ,r_{C}\right)$ to the region of $R^{2+}$, which is not excluded by theoretical arguments [30] or experimental bounds [24,25]. With good approximation, $D_{\lambda ,r_{C}}$ can be parameterized as follows:$$\lambda \geq \frac{10^{-30.2}}{r_{C}^{2}}=\frac{a_{1}}{r_{C}^{2}};\;\lambda \leq \frac{10^{-22.4}}{r_{C}^{2}}=\frac{a_{2}}{r_{C}^{2}}$$
(8)$$\lambda \geq 10^{-9.6}\;\;r_{C}^{2}=a_{3}\;r_{C}^{2};\;\lambda \leq 10^{2.8}\;\;r_{C}^{2}=a_{4}\;r_{C}^{2}.$$

The domain $D_{\lambda ,r_{C}}$ is shown in Figure 1, and, accordingly, the prior is expressed as product of Heaviside functions:(9)$${\tilde{p}}_{0}\left(\lambda ,r_{C}\right)=\vartheta \left(\lambda -\frac{a_{1}}{r_{C}^{2}}\right)\cdot \vartheta \left(\frac{a_{2}}{r_{C}^{2}}-\lambda \right)\cdot \vartheta \left(\lambda -a_{3}r_{C}^{2}\right)\cdot \vartheta \left(a_{4}r_{C}^{2}-\lambda \right).$$

The joint *pdf* is represented in Figure 2.

### 3.1. *pdf* of λ

The *pdf* of λ is obtained by marginalizing the joint *pdf* over $r_{C}$. As shown in Figure 1, given the domain $D_{\lambda ,r_{C}}$, the functional dependence $r_{C}=r_{C}\left(\lambda \right)$ changes in different intervals of the λ domain. For this reason, the *pdf* of λ is a piecewise-defined function given by the following relations: (10)$$\tilde{p}\left(\lambda \right)=\left\{\begin{array}{cc} {\tilde{p}}_{1}\left(\lambda \right)=\frac{1}{N}\int_{\sqrt{a_{1}/\lambda }}^{\sqrt{\lambda /a_{3}}}{\left(a\frac{\lambda }{r_{c}^{2}}+\Lambda_{b}+1\right)}^{z_{c}}e^{-\left(a\frac{\lambda }{r_{c}^{2}}+\Lambda_{b}+1\right)}dr_{c} & \;\lambda_{\min}\geq \lambda >\lambda_{1}\\ {\tilde{p}}_{2}\left(\lambda \right)=\frac{1}{N}\int_{\sqrt{a_{1}/\lambda }}^{\sqrt{a_{2}/\lambda }}{\left(a\frac{\lambda }{r_{c}^{2}}+\Lambda_{b}+1\right)}^{z_{c}}e^{-\left(a\frac{\lambda }{r_{c}^{2}}+\Lambda_{b}+1\right)}dr_{c} & \;\lambda_{1}\geq \lambda >\lambda_{2}\\ {\tilde{p}}_{3}\left(\lambda \right)=\frac{1}{N}\int_{\sqrt{\lambda /a_{4}}}^{\sqrt{a_{2}/\lambda }}{\left(a\frac{\lambda }{r_{c}^{2}}+\Lambda_{b}+1\right)}^{z_{c}}e^{-\left(a\frac{\lambda }{r_{c}^{2}}+\Lambda_{b}+1\right)}dr_{c} & \;\lambda_{2}\geq \lambda >\lambda_{\max} \end{array}\right.$$
where $\lambda_{\min}$ and $\lambda_{\max}$ are defined by the conditions
$$\left\{\begin{array}{c} \lambda /a_{3}=a_{1}/\lambda \;\;\Rightarrow \;\;\lambda_{\min}=\sqrt{a_{1}a_{3}}\\ \lambda /a_{4}=a_{2}/\lambda \;\;\Rightarrow \;\;\lambda_{\max}=\sqrt{a_{2}a_{4}} \end{array}\right.,$$

$\lambda_{1}$ and $\lambda_{2}$ by
$$\left\{\begin{array}{c} \lambda /a_{3}=a_{2}/\lambda \;\;\Rightarrow \;\;\lambda_{1}=\sqrt{a_{2}a_{3}}\\ \lambda /a_{4}=a_{1}/\lambda \;\;\Rightarrow \;\;\lambda_{2}=\sqrt{a_{1}a_{4}} \end{array}\right.,$$
and N is a normalization constant. Let us consider the generic integral $M_{i}\left(\lambda \right)$
(11)$$M_{i}\left(\lambda \right)=\int_{l_{1}}^{l_{2}}{\left(a\frac{\lambda }{r_{C}^{2}}+\Lambda_{b}+1\right)}^{z_{c}}e^{-\left(a\frac{\lambda }{r_{C}^{2}}+\Lambda_{b}+1\right)}dr_{C}\;.$$

By applying the variable transformation
(12)$$\frac{a\;\lambda }{r_{C}^{2}}=\xi \;;\;d\xi =\frac{-2}{{\left(a\lambda \right)}^{1/2}}\xi^{3/2}dr_{C}\;,$$

Equation (Equation 11) can be rewritten as
(13)$$M_{i}\left(\lambda \right)=\int_{\frac{a\lambda }{l_{1}^{2}}}^{\frac{a\lambda }{l_{2}^{2}}}{\left(\xi +\Lambda_{b}+1\right)}^{z_{c}}e^{-\left(\xi +\Lambda_{b}+1\right)}\left(-\frac{{\left(a\lambda \right)}^{1/2}}{2}\right)\xi^{-\frac{3}{2}}d\xi .$$

A binomial expansion of the term ${\left(\xi +\Lambda_{b}+1\right)}^{z_{c}}$ yields
Mi(λ)=∑k=0zczck−(aλ)1/22e−Λb+1Λb+1zc−k∫aλl12aλl22ξk−32e−ξdξ=
(14)=∑k=0zczck−(aλ)1/22e−Λb+1Λb+1zc−k·γk−12,aλl22−γk−12,aλl12,
where γ represents the lower incomplete gamma function. Finally, the normalization N is given by
(15)$$N=\int_{\lambda_{\min}}^{\lambda_{1}}M_{1}\left(\lambda \right)\;d\lambda +\int_{\lambda_{1}}^{\lambda_{2}}M_{2}\left(\lambda \right)\;d\lambda +\int_{\lambda_{2}}^{\lambda_{\max}}M_{3}\left(\lambda \right)\;d\lambda .$$

It is easy to check that $\tilde{p}\left(\lambda \right)$ is a continuous function of λ, shown in Figure 3. The cusp at $\lambda ∼10^{-16}$ s${}^{-1}$ is a consequence of the marginalization in $r_{C}$ and corresponds to the edge of the $r_{C}$ domain. Although the cusp coincides with the global maximum of $\tilde{p}\left(\lambda \right)$, the probability that λ is less than $10^{-13}$ is negligible. That can be checked by comparison with the cumulative distribution $\tilde{P}\left(\lambda \right)$, represented in Figure 4. It is worth noticing that $\tilde{p}\left(\lambda \right)$ does not represent a *measurement* of λ; the spontaneous radiation yield is proportional to $\lambda /r_{C}^{2}$, and no evidence of collapse signal can be inferred from the λ distribution alone.

Indeed, ongoing and future more sensitive radiation or gravitational wave measurements will affect $\tilde{p}\left(\lambda \right)$ by shifting the pdf downward in λ. However, $\tilde{P}\left(\lambda \right)$ gathers rich statistical information and allows setting consistent upper bounds on λ alone, exploiting the available experimental and theoretical knowledge.

### 3.2. *pdf* of $r_{C}$

The *pdf* of $r_{C}$ is obtained by marginalizing the joint *pdf* over λ. Depending on the $r_{C}$ intervals, shown in Figure 1, the functional dependence $\lambda =\lambda (r_{C})$ changes, and consequently, the *pdf* of $r_{C}$ is piecewise-defined as follows: (16)$$\tilde{p}\left(r_{C}\right)=\left\{\begin{array}{cc} {\tilde{p}}_{1}\left(r_{C}\right)=\frac{1}{N}\int_{\frac{a_{1}}{r_{C}^{2}}}^{a_{4}r_{C}^{2}}{\left(a\frac{\lambda }{r_{c}^{2}}+\Lambda_{b}+1\right)}^{z_{c}}e^{-\left(a\frac{\lambda }{r_{c}^{2}}+\Lambda_{b}+1\right)}d\lambda & \;r_{C,\min}<r_{C}<\alpha_{1}\\ {\tilde{p}}_{2}\left(r_{C}\right)=\frac{1}{N}\int_{\frac{a_{1}}{r_{C}^{2}}}^{\frac{a_{2}}{r_{C}^{2}}}{\left(a\frac{\lambda }{r_{c}^{2}}+\Lambda_{b}+1\right)}^{z_{c}}e^{-\left(a\frac{\lambda }{r_{c}^{2}}+\Lambda_{b}+1\right)}d\lambda & \;\alpha_{1}<r_{C}<\alpha_{2}\\ {\tilde{p}}_{3}\left(r_{C}\right)=\frac{1}{N}\int_{a_{3}r_{C}^{2}}^{\frac{a_{2}}{r_{C}^{2}}}{\left(a\frac{\lambda }{r_{c}^{2}}+\Lambda_{b}+1\right)}^{z_{c}}e^{-\left(a\frac{\lambda }{r_{c}^{2}}+\Lambda_{b}+1\right)}d\lambda & \;\alpha_{2}<r_{C}<r_{C,\max} \end{array}\right.,$$
with
$$\left\{\begin{array}{c} r_{C,\min}={\left(a_{1}/a_{4}\right)}^{\frac{1}{4}}\\ \alpha_{1}={\left(a_{2}/a_{4}\right)}^{\frac{1}{4}}\\ \alpha_{2}={\left(a_{1}/a_{3}\right)}^{\frac{1}{4}}\\ r_{C,\max}={\left(a_{2}/a_{3}\right)}^{\frac{1}{4}}\;. \end{array}\right.,$$

The integral to be solved in the generic $r_{C}$ range is
$$N_{i}\left(r_{C}\right)=\int_{l_{1}}^{l_{2}}{\left(a\frac{\lambda }{r_{C}^{2}}+\Lambda_{b}+1\right)}^{z_{c}}e^{-\left(a\frac{\lambda }{r_{C}^{2}}+\Lambda_{b}+1\right)}d\lambda =$$
$$=\int_{\frac{a\;l_{1}}{r_{C}^{2}}+\Lambda_{b}+1}^{\frac{a\;l_{2}}{r_{C}^{2}}+\Lambda_{b}+1}\xi^{z_{c}}\;e^{-\xi }\;\frac{r_{C}^{2}}{a}d\xi =$$
(17)$$=\frac{r_{C}^{2}}{a}\left[\gamma \left(z_{c}+1,\frac{a\;l_{2}}{r_{C}^{2}}+\Lambda_{b}+1\right)-\gamma \left(z_{c}+1,\frac{a\;l_{1}}{r_{C}^{2}}+\Lambda_{b}+1\right)\right].$$

From Equation (Equation 17), it can be verified that $\tilde{p}\left(r_{C}\right)$ is a continuous function. Hence, the normalization is given by: (18)$$N=\int_{r_{C,\min}}^{\alpha_{1}}N_{1}\left(r_{C}\right)\;dr_{C}+\int_{\alpha_{1}}^{\alpha_{2}}N_{2}\left(r_{C}\right)\;dr_{C}+\int_{\alpha_{2}}^{r_{C,\max}}N_{3}\left(r_{C}\right)\;dr_{C}.$$

$\tilde{p}\left(r_{C}\right)$ is shown in Figure 5. The marginal distribution contains relevant information. The *pdf* is narrow around the mode at $r_{C}∼10^{-6}$ m. Let us point out again that this has not to be interpreted as a *measurement* of $r_{C}$. Rather, $\tilde{p}\left(r_{C}\right)$ implies that, up to all the available experimental and theoretical knowledge, $r_{C}$ values smaller than $2\cdot 10^{-7}$ m are unlikely. That can be checked using the corresponding cumulative distribution $\tilde{P}\left(r_{C}\right)$, shown in Figure 6.

Ongoing and forthcoming more sensitive measurements of spontaneous radiation (see, e.g., [28,35]) will trigger sizably smaller $a_{4}$ values, thus moving the mode of $\tilde{p}\left(r_{C}\right)$ towards greater $r_{C}$ values. Indeed, the expected spontaneous radiation yield in Equation (Equation 1) diminishes with the growing $r_{C}$. On the other hand, a smaller $a_{2}$ is foreseen in Ref. [24] as a consequence of an improved force noise in the LISA Pathfinder experiment. Considering that, in this case, the force noise spectral density is expected to increase with $r_{C}$, this would slightly decrease the mode of $\tilde{p}\left(r_{C}\right)$. Therefore, in view of the simultaneous $a_{4}$ improvement, this would not significantly impact the results of this analysis.

## 4. Comparison with the Analysis in Terms of Exclusion Region

Previous spontaneous radiation measurements aimed to extract limits on the ratio $\lambda /r_{C}^{2}$, as mentioned in Section 1. That was accomplished, e.g., in Ref. [27], by extracting the pdf of the stochastic variable $\lambda /r_{C}^{2}$ and equating the cumulative to a probability Π=0.95. Since individual prior information on λ or $r_{C}$ could not be implemented, a uniform prior was adopted for $\lambda /r_{C}^{2}$.

As a consistency check, let us use the joint pdf$\tilde{p}\left(\lambda ,r_{C}\right)$ to find the value of the parameter μ, which corresponds to the exclusion region $\lambda <\mu \;r_{C}^{2}$ in the ($\lambda ,r_{C}$) plane, consistent with the analysis in Ref. [27]. To this end, the priors are assumed uniform over the domain:$$\left\{\begin{array}{c} 0<\lambda <\mu r_{C}^{2}\\ r_{C1}<r_{C}<r_{C2} \end{array}\right.$$
and zero outside. An arbitrarily big upper limit on $r_{C}$ must be set to make the *pdf* normalizable. The lower limit $r_{C1}$ is introduced to avoid the pole in $r_{C}=0$. We then have to solve the integral equation for μ: (19)$$\frac{\int_{r_{C1}}^{r_{C2}}dr_{C}\int_{0}^{\mu r_{C}^{2}}{\left(a\frac{\lambda }{r_{C}^{2}}+\Lambda_{b}+1\right)}^{z_{c}}e^{-\left(a\frac{\lambda }{r_{C}^{2}}+\Lambda_{b}+1\right)}d\lambda }{N^{\prime }}=\Pi ,$$
with the normalization
(20)$$N^{\prime }=\int_{r_{C1}}^{r_{C2}}dr_{C}\int_{0}^{\infty }{\left(a\frac{\lambda }{r_{C}^{2}}+\Lambda_{b}+1\right)}^{z_{c}}e^{-\left(a\frac{\lambda }{r_{C}^{2}}+\Lambda_{b}+1\right)}d\lambda .$$

That yields the equation
(21)$$\frac{\gamma \left(z_{c}+1,a\mu +\Lambda_{b}+1\right)-\gamma \left(z_{c}+1,\Lambda_{b}+1\right)}{\Gamma \left(z_{c}+1,\Lambda_{b}+1\right)}=\Pi ,$$
whose solution is, as expected, μ=52 for Π=0.95. The result obtained in Ref. [27] is recovered, demonstrating the correctness of the approach.

## 5. Discussion, Conclusions, and Perspectives

In this work, a new methodology is proposed for interpreting data from spontaneous radiation search experiments in the context of the CSL model. So far, these studies have aimed to extract the upper bounds on the ratio $\lambda /r_{C}^{2}$ of the two CSL parameters, which is proportional to the expected spontaneous emission rate. When the whole prior information is exploited from previous experimental and theoretical bounds, the joint *pdf* $\tilde{p}\left(\lambda ,r_{C}\right)$ can be calculated, and the marginalized posteriors $\tilde{p}\left(\lambda \right)$ and $\tilde{p}\left(r_{C}\right)$ can be obtained. An example of application of this methodology is given, by exploiting the data collected in Ref. [27], which also provides the absolute background estimate input propaedeutical to the Bayesian inference. $\tilde{p}\left(\lambda \right)$ and $\tilde{p}\left(r_{C}\right)$ are found to contain valuable statistical information. In particular, $r_{C}$ results in a greater than the originally proposed value of $10^{-7}$ m [3] with a probability close to 1. As a crosscheck, the previous bound on $\lambda /r_{C}^{2}$ is recovered using the joint $\tilde{p}\left(\lambda ,r_{C}\right)$ when the priors are relaxed to mimic the previous *one-parameter* analyses.

It is worth noticing that the methodology outlined in this work is particularly interesting for analyzing non-Markovian generalizations of the CSL [36,37,38] and other models of dynamical wave function collapse. Non-Markovianity would imply a modification of the priors definitions and require the introduction of new phenomenological parameters in the models (e.g., a cutoff frequency), making this approach extremely appealing.

A systematic study of data from ongoing [28,35] and forthcoming experiments, in analogy with what is presented in this article, would supplement our conclusions and push further the limit on the correlation length.

Our group is implementing a new experimental setup based on cutting-edge Germanium detectors and a refined numerical implementation of the presented methodology. These are addressed directly to exploit the strong energy dependence features expected for non-Markovian implementations of the current collapse models, which stand out as consistent solutions to the quantum-to-classical transition conundrum and the related measurement problem.

## Figures and Tables

**Figure 1 entropy-25-00295-f001:**
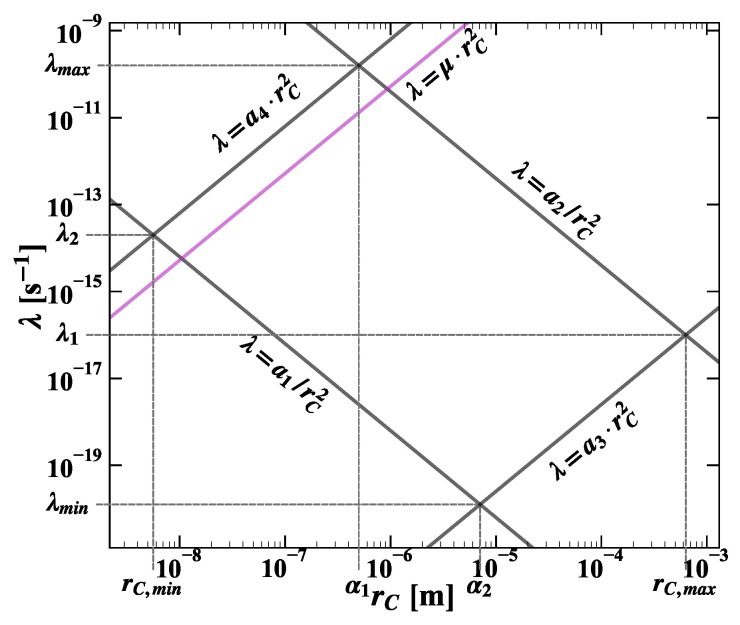
In the figure, the joint *pdf* domain $\tilde{p}\left(\lambda ,r_{C}\right)$ is shown as defined by the parameterization described in Equation (Equation 8) of the text.

**Figure 2 entropy-25-00295-f002:**
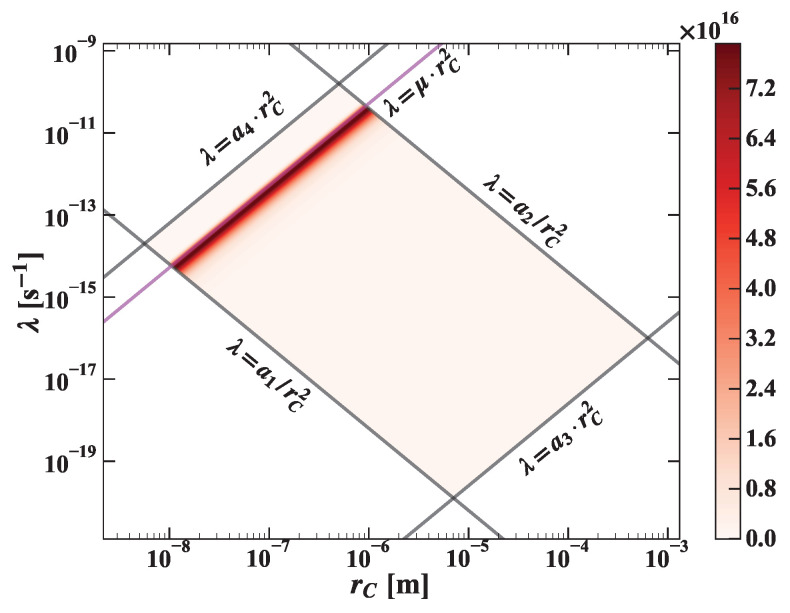
The joint *pdf* $\tilde{p}\left(\lambda ,r_{C}\right)$ is shown in the figure.

**Figure 3 entropy-25-00295-f003:**
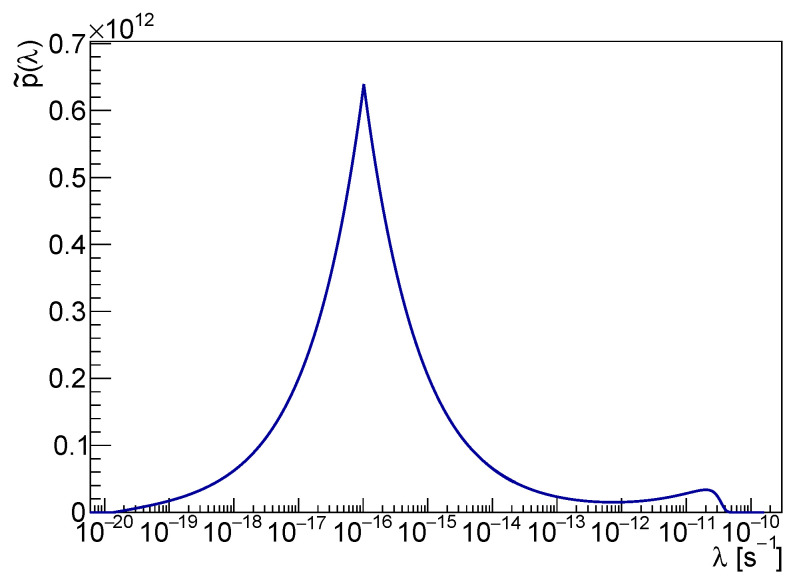
The *pdf* of λ in the logarithmic scale is shown in the figure.

**Figure 4 entropy-25-00295-f004:**
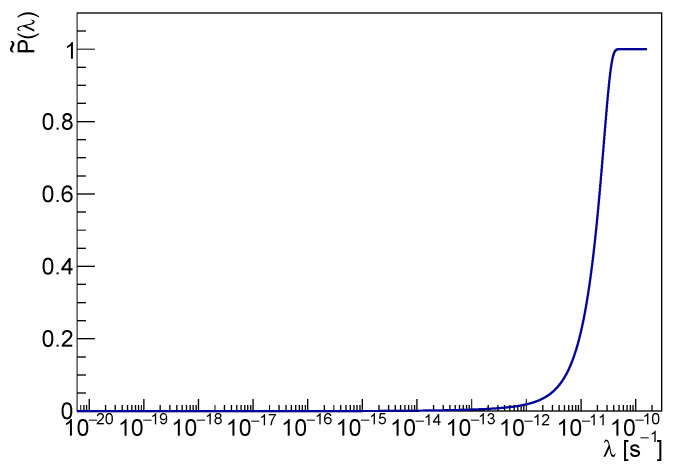
The cumulative *pdf* of λ in the logarithmic scale is shown in the figure.

**Figure 5 entropy-25-00295-f005:**
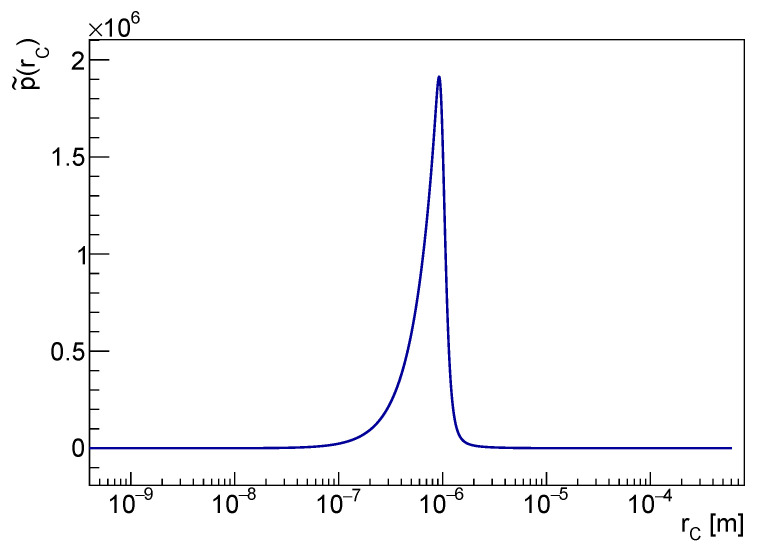
The *pdf* of $r_{C}$ in the logarithmic scale is shown in the figure.

**Figure 6 entropy-25-00295-f006:**
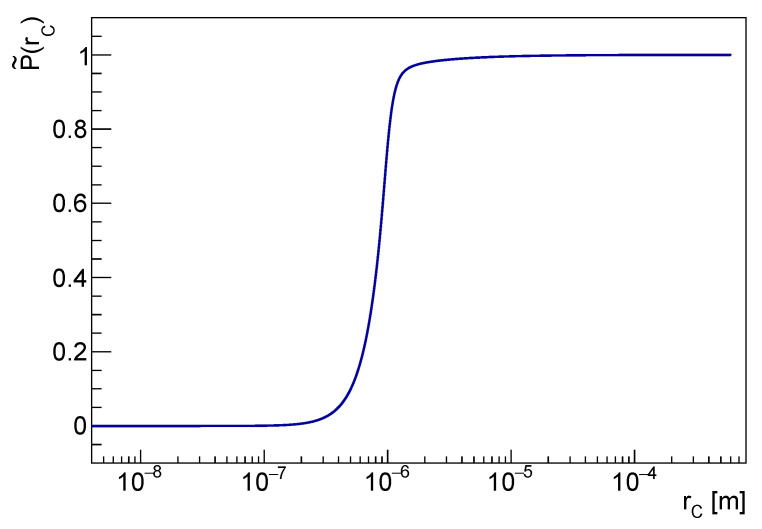
The cumulative *pdf* of $r_{C}$ in the logarithmic scale is shown in the figure.

## Data Availability

The data presented in this study are available on request from the corresponding author.

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
