# Peer review of "A Novel Approach to Parameter Determination of the Continuous Spontaneous Localization Collapse Model"

_entropy, 2023, doi:10.3390/e25020295_

Round 1

Reviewer 1 Report

The manuscript proposes a novel method for employing data from spontaneous radiation search experiments to determine the CSL model parameters. While previous work used such data to impose upper bounds on a particular combination of the strength and correlation length parameters, this manuscript employs prior theoretical and experimental information to calculate marginalized posteriors for such parameters. These marginalized posteriors are argued to disclose a deeper insight into the nature of the CSL model.

I find the proposed method useful and I agree that it allows for a more profound exploration of the parameters space of the CSL model. The only question I have has to do with the fact that the pdf for \lambda contains a cusp, as well as with the comment on the issue offered in the paper. It is argued that the cusp corresponds to the edge of the correlation length domain and that it gives no valuable statistical information. The issue is that such a domain is precisely what contains the prior information being employed to go beyond previous analyses. It seems strange, then, to argue, both, that this method allows for a deeper statistical understanding and that some features of the result obtained contain no valuable statistical information. A more nuanced discussion of what aspects of the analysis offered are to be trusted and which not would be welcome.

Minor points:

Line 25: it is not clear what “argument” is being alluded.

Line 29: The “collapsing” field has not been defined.

Line 105: \Lambda_c should be defined before or right after Eq. (2).

Line 110: The “hence” at the beginning of the line is confusing.

Line 124: There are typos in Eq. (7): the integral in the denominator of the first line is missing a “dr_c” and there is an extra arrow at the end of the first line.

Author Response

REVIEWER 1

«The manuscript proposes a novel method for employing data from spontaneous radiation search experiments to determine the CSL model parameters. While previous work used such data to impose upper bounds on a particular combination of the strength and correlation length parameters, this manuscript employs prior theoretical and experimental information to calculate marginalized posteriors for such parameters. These marginalized posteriors are argued to disclose a deeper insight into the nature of the CSL model.

I find the proposed method useful and I agree that it allows for a more profound exploration of the parameters space of the CSL model. The only question I have has to do with the fact that the pdf for \lambda contains a cusp, as well as with the comment on the issue offered in the paper. It is argued that the cusp corresponds to the edge of the correlation length domain and that it gives no valuable statistical information. The issue is that such a domain is precisely what contains the prior information being employed to go beyond previous analyses. It seems strange, then, to argue, both, that this method allows for a deeper statistical understanding and that some features of the result obtained contain no valuable statistical information. A more nuanced discussion of what aspects of the analysis offered are to be trusted and which not would be welcome.»

We thank the Reviewer for our paper's in-depth and careful review and, in particular, for his comments and suggestions. All comments and suggestions were addressed in the revised version of the manuscript, which improved the quality of the paper.

More in detail, we thank the Reviewer for the question on the cusp, which appears in the pdf for \lambda, since the corresponding sentence in the manuscript was indeed misleading. We did not mean that the edge of the correlation length does not give important statistical information. As the Reviewer correctly points out, the domain – i.e., the prior – is a crucial ingredient of this analysis. The cusp appears due to the edge when integrating over $r_C$, but although it corresponds to a global maximum of the $\lambda$ pdf, the probability that $\lambda$ belongs to the region of the cusp is negligible. Please note that since the scale in Figure 3 is logarithmic, the integral of the pdf in the range ($10^{-20}$-$10^{-13}$) s$^{-1}$ is negligible. That is more evident by comparison with the cumulative $\lambda$ pdf distribution shown in Figure 4.

To better clarify this important point, the corresponding sentence in the text was rephrased as follows:

“The cusp at $\lambda\sim 10^{-16}$ s$^{-1}$ is a consequence of the marginalization in $r_C$ and corresponds to the edge of the $r_C$ domain. Although the cusp coincides with the global maximum of $\tilde{p}(\lambda)$, the probability that $\lambda$ is less than $10^{-13}$ is negligible. That can be checked by comparison with the cumulative distribution $\tilde{P}(\lambda)$, represented in Figure \ref{cuml}." 

We also thank the Reviewer for his comments; all of them were addressed and clarified in the revised version of the manuscript as described below:

 «Minor points:

Line 25: it is not clear what argument” is being alluded.»

The word “argument” was replaced with “subject”

«Line 29: The collapsing” field has not been defined.»

The stochastic noise's physical origin, which induces the collapse in the Continuous Spontaneous Localization model, is still unknown. That is why we generically refer to a "collapsing" field.

«Line 105: \Lambda_c should be defined before or right after Eq. (2).»

We thank the Reviewer, in the revised version of the manuscript, we modified the sentence right after Eq. (2) as follows:

"the expected number of total counts ($\Lambda_c$) can be expressed in terms of the expected signal ($\Lambda_s$) and background ($\Lambda_b$) contributions:"

«Line 110: The hence” at the beginning of the line is confusing.»

We replaced “hence” with “and” after Eq. (4).

«Line 124: There are typos in Eq. (7): the integral in the denominator of the first line is missing a dr_c” and there is an extra arrow at the end of the first line.»

We thank the Reviewer for having noticed the typos. We corrected the formula accordingly.

Accordingly, to the Reviewer's suggestion, we revised the English extensively to correct grammar, language, and style. We thank the Reviewer one more time for the sedulous redaction.

Reviewer 2 Report

In the present manuscript the authors introduce a novel approach to establish empirical bounds on the parameters of the so called Continuous Spontaneous Localization (CSL) model.  The CSL model aims to provide a solution to the quantum measurement problem by introducing non-linear and stochastic modifications to the Schroedinger equation. These modifications are characterized by phenomenological parameters, also called the "collapse parameters", i.e. the "strength" lambda and the correlation length r_c .  Traditionally, laboratory experiments have put bounds on the parameter space lambda--r_c.  The present article is devoted to introduce a method for disentangling the probability distribution function (pdf) of the two parameters.

After reviewing the paper I think the subject of this work is worth exploring, the hypotheses regarding the priors are presented clearly and the results obtained deserved to be published.  However, before I recommend this manuscript for publication, I require the following points to be clarified :  

1. In Eq. (7) the denominator is integrated over r_c and lambda. However, in the unnumbered equation of the previous line, the integral appearing in the denominator is only integrated over lambda. I think there is a typo,  i.e. the integration over r_c is missing in such an equation.

2.  In most of the analyses employing Bayesian methods, after finding the posterior (or likelihood),  it is customary to obtain the maximum likelihood estimators. These are the values of the parameters such that the likelihood function is maximized.  I think the authors can directly obtain such estimators from Eq. (7).  However, if the authors consider that this analysis is not necessary, then I would like them to include a brief discussion about this issue.

3. Related to the previous item, after finding the normalized posterior, one could also derive the regions of confidence for the parameters. Furthermore, in the particular case of the present paper, there are only two parameters, so calculating the regions of confidence is not very computing-expensive. Is there a reason why the authors consider this particular analysis not useful  and instead prefer to obtain only the cumulative pdf marginalized of each parameter?

4. I'm having trouble understanding the vertical axis of Figs. 3 and 5. As far as I understood,  p̃(lambda) and p̃(r_c) are the marginalized pdf for the parameters, and they were also normalized, i.e. these Figs. correspond to Eqs. (10) and (16) respectively. So I do not understand why the values on the vertical axis in each figure are not less than one, and instead they are of order 10^12 for p̃(lambda)  and 10^6 for p̃(r_c).

5.  Finally, I would like to ask the authors to comment in their paper whether they believe that this Bayesian analysis can be extended to other collapse models in which the noise field (which modifies the standard Schroedinger equation) is not of Brownian type. Can the type of noise affect (or not) the priors used?

Author Response

«In the present manuscript the authors introduce a novel approach to establish empirical bounds on the parameters of the so called Continuous Spontaneous Localization (CSL) model.  The CSL model aims to provide a solution to the quantum measurement problem by introducing non-linear and stochastic modifications to the Schroedinger equation. These modifications are characterized by phenomenological parameters, also called the "collapse parameters", i.e. the "strength" lambda and the correlation length r_c .  Traditionally, laboratory experiments have put bounds on the parameter space lambda--r_c.  The present article is devoted to introduce a method for disentangling the probability distribution function (pdf) of the two parameters.

After reviewing the paper I think the subject of this work is worth exploring, the hypotheses regarding the priors are presented clearly and the results obtained deserved to be published.  However, before I recommend this manuscript for publication, I require the following points to be clarified :»

We thank the Reviewer for the in-depth and careful review and his comments and suggestions. All of them were addressed in the manuscript's revised version and helped improve the quality of the paper.

Answers to the Reviewer's questions and a description of the modifications applied to the manuscript are given below:

«1. In Eq. (7) the denominator is integrated over r_c and lambda. However, in the unnumbered equation of the previous line, the integral appearing in the denominator is only integrated over lambda. I think there is a typo,  i.e. the integration over r_c is missing in such an equation.»

We thank the Reviewer for noticing this typo; the integration over dr_c was added in the first line of Eq. (7).

«2.  In most of the analyses employing Bayesian methods, after finding the posterior (or likelihood),  it is customary to obtain the maximum likelihood estimators. These are the values of the parameters such that the likelihood function is maximized. I think the authors can directly obtain such estimators from Eq. (7).  However, if the authors consider that this analysis is not necessary, then I would like them to include a brief discussion about this issue.»

We thank the Reviewer for addressing this important question. As stated in line 147 of the original manuscript version, spontaneous radiation is the physical observable of our (and similar) experiments studying wave function collapse models. However, spontaneous radiation searches still did not result in a clear detection of the collapse signal. Consequently, only upper limits on the signal rate (i.e., on the ratio $\lambda/r_C^2$) were set till now. In a previous work (Ref [27] of the manuscript), we extracted the pdf of $\lambda/r_C^2$, from which we estimated the resulting limit. In this sense, we think that the improved result of this analysis, namely the extraction of the individual pdfs of the two parameters, results in the possibility to set separate limits, upper limits on $lambda$ independently from the $r_C$ value or, the other way round, lower limits on r_C independently from the value of $\lambda$. The independence of the two limits is granted by the marginalization of the posterior, which washes away the dependence on the other variable.

That is why instead of maximum likelihood estimators or confidence regions for the two parameters, we preferred to present the cumulative marginalized pdf distributions, from which the reader can easily obtain limits corresponding to the desired probability values.

«3. Related to the previous item, after finding the normalized posterior, one could also derive the regions of confidence for the parameters. Furthermore, in the particular case of the present paper, there are only two parameters, so calculating the regions of confidence is not very computing-expensive. Is there a reason why the authors consider this particular analysis not useful  and instead prefer to obtain only the cumulative pdf marginalized of each parameter?»

We kindly address the Reviewer to the answer to the previous question, in which an answer to this specific question is also given.

«4. I'm having trouble understanding the vertical axis of Figs. 3 and 5. As far as I understood,  p̃(lambda) and p̃(r_c) are the marginalized pdf for the parameters, and they were also normalized, i.e. these Figs. correspond to Eqs. (10) and (16) respectively. So I do not understand why the values on the vertical axis in each figure are not less than one, and instead they are of order 10^12 for p̃(lambda)  and 10^6 for p̃(r_c).»

The Reviewer is right; p̃ (lambda) and p̃(r_c) are marginalized pdfs for the parameters, respectively Eqs. (10) and (16), and, as so, they are normalized to unity. The vertical axes extend to big numerical values (of the order of 10^12 for p̃(lambda)  and 10^6 for p̃(r_c)) because the corresponding ranges for which the probabilities of $\lambda$ and $r_C$ are sizably different from zero are respectively approximately $10^{-20} < \lambda < 4 \cdot 10^{-11}$ and $10^{-7} < r_C < 2 \cdot 10^{-6}$, therefore the value of the integrals is 1.

«5.  Finally, I would like to ask the authors to comment in their paper whether they believe that this Bayesian analysis can be extended to other collapse models in which the noise field (which modifies the standard Schroedinger equation) is not of Brownian type. Can the type of noise affect (or not) the priors used?»

We thank the Reviewer for this interesting and relevant question, which addresses a very important future application of the analysis outlined in this work.

Indeed, this analysis is also suitable for non-Markovian generalizations of the CSL and other collapse models that have been developed or are presently under study. As correctly pointed out by the Reviewer, the noise structure (e.g., the correlation in time) will modify the priors. Moreover, non-Markovian noise fields require further effective parameters in the models. Hence, the calculation of joint and marginalized posteriors will be extremely useful.

 In this context, we added the following sentence to Section 5, "Discussion, conclusions, and perspectives":

"It is worth noticing that the methodology outlined in this work is particularly interesting for analyzing non-Markovian generalizations of the CSL [36-38] and other models of dynamical wave function collapse. Non-Markovianity would imply a modification of the priors definitions and require the introduction of new phenomenological parameters in the models (e.g., a cutoff frequency), making this approach extremely appealing."